The spectacular human nose: an amplifier of individual quality?

Mikalsen Åse Kristine Rognmo 1
Folstad Ivar 1 ivar.folstad@uit.no
Yoccoz Nigel Gilles 1
Laeng Bruno 2 3
1 Department of Arctic and Marine Biology, University of Tromsø , Norway
2 Department of Psychology, University of Tromsø , Norway
Sorci Gabriele
3 Current affiliation: Department of Psychology, University of Oslo, Norway

Electronic publication date: 2014 Apr 17
Publication date: 2014
Volume: 2
Electronic Location ID: e357
Received 2014 Feb 12; Accepted 2014 Apr 1
Copyright: © 2014 Mikalsen et al.
Copyright year: 2014
Copyright holder: Mikalsen et al.
License: This is an open access article distributed under the terms of the Creative Commons Attribution License, which permits unrestricted use, distribution, and reproduction in any medium, provided the original author and source are credited.
License URL: https://creativecommons.org/licenses/by/3.0/

Keywords: Amplifiers, Signal, Human, Attractiveness, Nose, Asymmetry, Perception

Funding: The University of Tromsø financed the entire study The study was funded by the University of Tromsø. The funders had no role in study design, data collection and analysis, decision to publish, or preparation of the manuscript.

==============================
Amplifiers are signals that improve the perception of underlying differences in quality. They are cost free and advantageous to high quality individuals, but disadvantageous to low quality individuals, as poor quality is easier perceived because of the amplifier. For an amplifier to evolve, the average fitness benefit to the high quality individuals should be higher than the average cost for the low quality individuals. The human nose is, compared to the nose of most other primates, extraordinary large, fragile and easily broken—especially in male–male interactions. May it have evolved as an amplifier among high quality individuals, allowing easy assessment of individual quality and influencing the perception of attractiveness? We tested the latter by manipulating the position of the nose tip or, as a control, the mouth in facial pictures and had the pictures rated for attractiveness. Our results show that facial attractiveness failed to be influenced by mouth manipulations. Yet, facial attractiveness increased when the nose tip was artificially centered according to other facial features. Conversely, attractiveness decreased when the nose tip was displaced away from its central position. Our results suggest that our evaluation of attractiveness is clearly sensitive to the centering of the nose tip, possibly because it affects our perception of the face’s symmetry and/or averageness. However, whether such centering is related to individual quality remains unclear.

Introduction

Although there is no agreed upon definition of biological communication (Scott-Phillips, 2007), a signal may be defined as any act or structure that has evolved because it alters the behavior of other organisms and this response has co-evolved with signal evolution (Maynard-Smith & Harper, 2003). It is likely that no single mechanism can explain the enormous variety of signals. Instead, several mechanisms are likely to function simultaneously. There are at least three ways by which signals may be reliable, either by convention, by cost or by design (Hasson, 1990; Hasson, 1997). Signals reliable by convention are cost-free symbols and icons, and have evolved because there exists a mutual interest between signalers and recipients in information transfer (Silk, Kaldor & Boyd, 2000). Signals reliable by cost are costly to produce or maintain and the intensity of the signal strength is therefore proportional to the resources invested in the signal by the signaler. Consequently, high quality individuals holding large amounts of resources can afford more intense signals than low quality individuals (Zahavi, 1975). Signals reliable because of costs have evolved in situations where there is conflict of interest between sender and receiver (Hasson, 1997).

Signals reliable by design have also evolved under conflicts between sender and receiver entities (Hasson, 1997). Amplifiers, which are one group of design signals, have not been thoroughly recognized by researchers (Taylor, Hasson & Clark, 2000; Stenseth & Sætre, 2004, see however Ljetoff et al., 2007; Galvan & Sanz, 2008; Gualla, Cermelli & Castellano, 2008; Castellano & Cermelli, 2010). They are ideally cost-free signals and honest because their design improve the receiver’s ability to assess pre-existing differences in underlying quality between signalers. Only high quality signalers benefit from amplifiers, that is, the amplifier increases rewards for individuals of high quality because they increase the receiver’s ability to perceive real quality. Yet, at the same time poor quality is also perceived more easily (Hasson, 1989). The evolution of amplifiers depends on the frequency of high versus low quality signalers, and on whether all individuals display the trait or not. Additionally, the amplifier will only evolve if the average fitness benefit that the amplifier allele gives to its higher quality individuals is higher than the average cost, in fitness, to its lower quality carriers (Hasson, 1989).

One appropriate example of an amplifier might be the white tail feathers of males of several species of lekking birds (Fitzpatrick, 1998), e.g., the black grouse (Tetrao tetrix). These tail feathers, which are important for male attractiveness (Höglund et al., 1994), may be damaged in male–male fights at the lek. Additionally, the absence of melanin in white feathers weakens keratin and makes them particularly vulnerable to damage (Burtt, 1986; Bonser, 1995). Such damage is also easily perceived as breaks or aberrations of the white feathers stand out towards the background of the other dark tail feathers. Thus, white tail feathers of lekking males increase the females’, or the competing males’, ability to perceive competitive ability. This increases rewards only for individual males of high quality that are able to defend their vulnerable white tails. Yet, as all males develop the trait, low quality males are also easily recognized by aberrations in their tail feathers. In this study we question whether the protruding human nose could have evolved as an amplifier.

Homo erectus was probably the first hominid with a projecting nose (McKee, Poirier & McGraw, 2005) and although theories related to effective breathing in arid habitats may be invoked in explaining the evolution of the modern human nose (Glanville, 1969; Jones, Martin & Pilbeam, 1994; McKee, Poirier & McGraw, 2005), it is interesting to note that sexual selection has been considered important for the evolution of other conspicuous primate noses, e.g., that of the Proboscis monkey (Nasalas larvatus) and the colorful mandrill’s (Mandrillus spihinx) nose (Jones, Martin & Pilbeam, 1994; Dixson, 1998). The anatomical construction of the nose in humans is that of bone, with cartilage forming the tip, and it is much larger than the noses of most of the primate species. Although there are slight population differences in nose shapes and sizes (Hall & Hall, 1995; Comuzzie et al., 1995), there is also sexual dimorphism in nose size, with male’s noses on average being larger than female’s (Enlow, 1982; Genecov, Sinclair & Dechow, 1990; Hennessey, Kinsella & Waddington, 2002; Holton et al., 2014) and authors have suggested that the largest anatomical sexual dimorphism in the human face is represented by the protuberance of the nose, followed by the cheeks, as observed in 3-D (Burton, Bruce & Dench, 1993; Bruce & Young, 1998, see also O’Toole et al., 1997; Holton et al., 2014). Additionally, eye-tracking of humans has shown that, when observers are asked to quickly decide whether they are looking at a male or female face, gaze concentrates on a region near the base of the nose (Sæther et al., 2009).

The location of the nose in the middle region of the face draws much unconscious attention to it (Hsiao & Cottrell, 2008), and its exposed, protruding position and its physical qualities makes it particularly vulnerable. Indeed, the nose is one of the body parts most prone to damage (Rhee et al., 2004) and the commonest craniofacial fracture is that of the nasal bones. Moreover, most of the injuries of the nose are caused by interpersonal violence, mainly among males in fistfights (Hussain et al., 1994; Brink, 2009). Interestingly, fistfights seem to be a species-specific human behavior, and the human hand has been suggested to evolve through sexual selection for improved striking performance during hand-to-hand combat by males (Morgan & Carrier, 2013). Nose fractures from male–male aggression are also commonly seen as accidental injuries of sports activities (Bledsoe et al., 2006) but may also result from blows specifically aimed at the nasal region (Lessa & de Souza, 2006). Studies of facial attractiveness have also documented that the nose is important when assessing attractiveness (Jones, 1995) and by rating the attractiveness of several attributes of a face, before rating the entire face, Meerdink and coworkers (1990) found that attractiveness of the nose was highly correlated with overall attractiveness. Not surprisingly, among the plastic surgeries done for aesthetic reasons, a large amount of patients undergo rhinoplasty (Babuccu et al., 2003; Mondin, Rinaldo & Ferlito, 2005). In sum, the human nose has several attributes that resemble traits that might have evolved as amplifiers in other species.

Recently, Neby & Folstad (2013) showed that detection of asymmetry in ambiguous dot figures, vaguely resembling a human face, heavily relied on centrality of dots in the “nasal” region but not of centrality in the “mouth” or “eye” regions. This pattern was, however, only apparent when the observers initially associated the dot figures with a human face. Observers associating the dot figures with non-facial objects, such as a butterfly or a tree, showed no difference in sensitivity to vertical position of the decentralized dots. Yet, when the latter group of observers was instructed to see a human face in the dot figure they also became more sensitive to the dots in the “nasal” region (Neby & Folstad, 2013). These results provide indications that the brain may deal with information about facial asymmetry and averageness, i.e., attractiveness, heavily depending on the centrality in the nasal region.

The aim of this study was to evaluate one prediction derived from the hypothesis that the human nose may have evolved as an amplifier, that is, does the facial positioning of the nose tip have a particular influence on our evaluation of attractiveness? Although this prediction could also have been derived from another hypothesis of signal evolution (e.g., the handicap principle), a particular effect of nose tip position on attractiveness seems less likely expected from the more traditional physiological hypothesis (e.g., that the protruding nose evolved for retaining moisture). The influence of facial traits on attractiveness was examined by manipulating, on digital images of faces of young models of both sexes, the position of the nose tip or, as a control, that of another conspicuous and centrally-located feature of the face: the mouth. We assumed that the attractiveness ratings should be high for pictures with centered nose tips and low for those with skewed nose tips. Finally, we predicted that the centrality of the nose tip in relation to other facial features should be more important in attractiveness judgments than the centrality of the mouth.

Material and Methods

Manipulations

The pictures used for assessment were digital color images of student faces, with a width of 28.22 cm and a height of 21.17 cm, from Laeng’s laboratory at the Department of Psychology, University of Tromsø, Norway (see Fig. 1 for an example). The students had all approved the use of the pictures for studies such as the one presented here. The images consisted of 10 individual faces with a “neutral” expression (see Otta, Abrosio & Hoshino, 1996) seen in full front, with views of 5 males and 5 females, which were all Caucasian in their early to late twenties (mean 25.0, SD = 1.7 and mean 26.1, SD = 2.7, respectively). The pictures were edited in Adobe Photoshop version 7.0. Before manipulating the positions of the nose tip or the mouth, these traits were first made symmetrical. Symmetry was obtained by cutting the trait (nose or mouth) in half, and then this half-image was “flipped” and pasted over the other half of the character. The symmetric trait was thereafter centered exactly midway between the eyes along its original horizontal axis. By first creating a symmetric and centered nose or mouth before skewing them, any preexisting asymmetry or deviation in centerness in the traits of the models were removed.

Figure 1 A set of pictures used for attractiveness evaluation.

One set of pictures, in which the right side of the nose and the mouth were used to make the trait symmetric (A) unmanipulated face (B) right symmetric, centered mouth (C) right symmetric mouth, skewed 0.5 cm to the right (D) right symmetric mouth, skewed 1.0 cm to the right (E) right symmetric, centered nose (F) right symmetric nose, nose tip skewed 0.5 cm to the right (G) right symmetric nose, nose tip skewed 1.0 cm to the right.

To create a face with a non-centered trait, the mouth or the nose from the symmetric and centered picture was skewed towards one or the other side. For the nose skewing manipulations, only the fleshy nose tip was moved, without changing the position of the nose wings. This was done to simulate a break of the cartilage, which is the most vulnerable part (Hussain et al., 1994), and seems a conservative manipulation as nostril symmetry is likely to increase resolution of deviations in nose-tip centrality and vice versa. For the mouth skewing, the whole mouth was displaced. To skew each feature, the nudge function in Photoshop was used.

For each image either the nose or the mouth (but not both) were manipulated. The three main manipulations done were (i) the nose and mouth centered, (ii) the nose and mouth skewed 0.5 cm, and (iii) the nose and mouth skewed 1.0 cm. The extent of skewing seem to be well within the natural range of skewing in these characters, as evaluated from pictures of persons undergoing rhinoplasty (e.g., Jin et al., 2006). Additionally, even though they were not asked, a large number of the persons evaluating the pictures (see below) commented that they could not identify any visible differences between the pictures.

Each manipulation had four varieties of different “orientation”, (i) right-right oriented pictures, where the trait (either nose or mouth) was the mirror image of its right part, skewed right; (ii) right–left oriented, where the trait was the mirror image of its right part, skewed left; (iii) left–left oriented, where the trait was the mirror image of its left part, skewed left; (vi) left–right oriented, where the trait was the mirror image of its left part, skewed right. Thus, after the manipulations we had 24 manipulated images, and one original, unmanipulated picture. Half of the manipulated pictures consisted of nose manipulations, while the other half were mouth manipulations.

Evaluation

The pictures were printed on semi glossy picture paper using a laser color printer. They were then divided into sets of the four different orientations. Right–right oriented pictures of one person, left–right oriented pictures etc. There were 7 pictures in each set, since each set included an unmanipulated picture of the person, one picture with centered nose, one with nose tip skewed 0.5 cm, one with nose tip skewed 1.0 cm, one centered mouth picture, one with mouth skewed 0.5 cm and one with mouth skewed 1.0 cm. With 4 different types of orientation and 10 different models, this resulted in 40 sets of pictures. Each set of pictures had the manipulated character moved in only one direction. One woman and one man independently evaluated each set of pictures.

In order to standardize information transfer, participants evaluating the pictures were given an instruction sheet detailing how to evaluate the pictures, and they also had to report their age and sex. They were then given one set of pictures, in a random order, and ranked the 7 pictures for attractiveness (1 was assigned to least attractive; 7 assigned to the most attractive). They had no time limitations and none ever indicated that they identified manipulations in the images. 40 women and 40 men were enrolled as evaluators in the experiment (mean age = 24.39; SD = 6.03), and half of the participants were given same-sex pictures. Evaluators where all students at the University of Tromsø, and they were unaware of the purpose of the study. Each evaluator was rewarded with a scratch-to-win lottery ticket.

Statistics

The statistical program used was R version 2.5 (R Development Core Team, 2007). As data consisted of ranks, we first considered non-parametric tests. The Friedman rank sum test suits unreplicated blocked data, a block being here one set of pictures. We used it to assess the effect of the main variable, “type of manipulation”, both overall and for each level of the other factors. However, as this non-parametric test could not be used to assess interactions among type of manipulation and other factors (orientation, sex of individual viewer and sex of viewer), we considered a parametric analysis of variance (ANOVA) model. Given that the sum of ratings given by a subject was constant (and equal to 1 + 2 + ⋯ + 7 = 28), the assumption of independence was not fulfilled (we expect a negative correlation among observations from the same subject). Classical mixed-effects models with subject as a random factor are not adequate since they can only model positive correlation (Pinheiro & Bates, 2000). We therefore considered estimating equation models (Zeger & Liang, 1986) that allow for flexible correlation structure to assess interactions, and given that estimates and standard errors obtained were similar to those obtained using a simpler ANOVA model, decided to present only the latter. The ANOVA model had “attractiveness ratings” as response variable, “type of manipulation” (6 levels; unmanipulated was the reference level), “orientation” (4 levels; right-right as reference level), “sex of individual viewed” (2 levels: female/male), and “sex of viewer” (2 levels: female/male) as predictor variables. Coefficients in the model represent differences between levels and the reference level for each factor (i.e., contrasts). In particular, they were used to evaluate whether the different manipulations (“nose centered”, “nose tip skewed 0.5 cm”, “nose tip skewed 1.0 cm”, “mouth centered”, “mouth skewed 0.5 cm”, “mouth skewed 1.0 cm”) were different from the unmanipulated face. We used Dunnett’s test and associated confidence intervals to control for multiple testing as “unmanipulated face” was a unambiguous reference level.

Results

The Friedman rank sum test gave strong evidence against the null hypothesis of no effect of “type of manipulation” on attractiveness ratings (χ2 = 95.3, d.f. = 6, p < 0.0001). Considering each orientation separately, the evidence was as strong (all χ2 > 21, d.f. = 6, p < 0.002).

Tests for interactions between manipulation and sex of individual viewed (F7,518 = 0.44, p = 0.88), manipulation and sex of viewer (F7,518 = 0.58, p = 0.77) and manipulation and orientation (F21,518 = 1.18, p = 0.26) showed that only the effect of manipulation needed to be retained in the model (there were no strong evidence for higher order interactions either, all P > 0.06). The model with only manipulation as predictor variable showed the centered nose to be significantly more attractive than the other manipulations (Fig. 2A). The nose skewed 1.0 cm was the least attractive among the different manipulations, followed by the mouth skewed 1.0 cm. The untreated face was the second most attractive after the centered nose (Fig. 2A).

Figure 2 Result summary.

(A) Dunnett’s 95% confidence intervals for the difference between manipulation levels of Mouth and Nose and the reference level “Unmanipulated”. (B) Box plots of attractiveness ratings for the different manipulations (1 least attractive, 7 most attractive). The uppermost line of the box is the upper (75%) quartile, the lowest line of the box is the lower (25%) quartile and the line in the middle is the median. Box plots are based on average values calculated by orientation.

Figure 2B shows the distribution of the attractiveness ratings for the different types of manipulations. The attractiveness ratings followed a pattern close to the one predicted, that is, pictures with the centered nose were rated as the most attractive, followed by the unmanipulated face, the centered mouth, the nose skewed 0.5 cm, the mouth skewed 0.5 cm, the mouth skewed 1.0 cm, and last, the nose skewed 1.0 cm. The ratings were quite consistent as shown by the relatively small amount of variation of a given level of manipulation.

Discussion

Effects on perceived attractiveness were clearly found when the nose was the manipulated feature, while none were found for the mouth manipulations, despite these were of the same magnitude than that used for the nose. The most extreme, off-centered, deviation of the nose resulted in decreased perceived attractiveness, whereas an artificially centered nose resulted in increased perceived attractiveness. Finally, the unmanipulated face had higher attractiveness than all of the manipulated faces except for the one with the centered nose. The present results suggest that our sense of attractiveness is affected more by the position of the nose than the mouth. However, the present findings do not exclude the possibility that more extreme changes in mouth position than those used here could also result in decreased attractiveness. Moreover, more extreme changes than those used in the current study may pass the threshold of what is considered a normal (non-pathological) variation in facial asymmetry.

There was no difference in the attractiveness ratings of male and female viewers and these attractiveness ratings were not related to whether faces of one’s own or of the other sex were viewed. That is, males and females do not seem to disagree on ratings of either male or female faces under the current manipulations. This is contrary to what could have been anticipated if intrasexual competition resulting in nose destructions was more important among males, which also have the largest noses. Sex differences in facial perception have been found in studies with the eye tracking method (Mathisen, 2002). That is, both males and females tend to focus on the nose of males but they are more likely to focus on the mouth of females. One would expect that the nose in men would be the focus of attention more often than for women, as a way to assess how they cope with a world that is neither soft nor flat, and where reduced social skills and reduced competitive ability also might result in a broken and skewed nose. However, both the nose and the mouth might also be sex hormone markers (Thornhill & Gangestad, 1996) and masculinization of upper face height (i.e., elongation of distance from anterior nasal spine to nasion) is seen in boys with delayed puberty when administered testosterone (Verdonck et al., 1999). Thus, testosterone dependent traits like the nose, may be a focus of attention since they also reveal underlying qualities not related to behavioral abilities in a multidimensional landscape (Folstad & Karter, 1992).

Movements of centered facial features may influence both the perceived symmetry (Neby & Folstad, 2013) and averageness, which both independantly are of importance for facial attractiveness (Rhodes, 2006). Yet, as slight deviations from symmetry and averageness can have little effect on evaluations of attractiveness (Langlois, Roggmann & Musselman, 1994), this might explain why there was no effect, in the present study, on attractiveness of the small positional manipulations. However, note that large movements of the mouth did not influence attractiveness, but there was an effect of the nose tip manipulations. The latter occurred both when the nose was centered and when it was skewed the most, and in both cases the effect was in the predicted direction, that is, more attractive when centered and less attractive when moved off-centre. The difference in the effects of manipulations of the mouth and the nose hints to their different importance for perception of symmetry or averageness. The nose is a relative static construction compared to the mouth (teeth excluded). It is not muscularized to the same extent as the mouth, which is heavily employed in both facial expression and speech (Fridlund, 1994). The nose is also closer to the centre of gravity of the face when it is viewed in full frontal orientation and when the tip is moved away from the centre position it makes the cartilage appear distorted from the nasal ridge. Given that the nasal ridge represents a vertical axis for evaluation of facial symmetry, its orientation may have a large influence on perception of symmetry (Evans, Wenderoth & Cheng, 2000). For example, when the nose tip is off-centered, one half of the face may appear larger than the other. This can negatively affect attractiveness (Baudoin & Tiberghien, 2004). A centered nose, on the other hand, will make each side of the face appear similar in size and might have the opposite effect. Additionally, an off-centered nose may also decrease averageness and increase distinctiveness, which both independently may be associated with reduced attractiveness (O’Toole et al., 1998; Little & Hancock, 2002). The opposite effect might explain the increased attractiveness of a centered nose. In sum, humans may not have developed adaptations towards perceiving off-centering in mouth position, but may to a larger extent perceive nose tip position when evaluating facial attractiveness. Whether this bias in perception is related to averageness or symmetry or both has yet to be established, but one can surmise that manipulations of the nose, which is a centrally-located and conspicuous physical trait, might have a particular strong effect on perception of symmetry (see Neby & Folstad, 2013). It is likely that slight deviations from average would be difficult to be noticed perceptually and consequently they would have negligible effects on our sense of attractiveness of a face, whereas slight deviations from symmetry might be more easily detected and, in turn, they might carry a larger weight on attractiveness judgments. Indeed, psychophysics studies on the perception of symmetry of visual forms have shown that the symmetry of a pattern can be judged at a very early stage in visual processing (Brooks & van der Zwan, 2002) and it is likely to be an effortless task. In contrast, the assessment of whether a specific pattern or array is equal or differs from the average of a class (e.g., in size) requires attention (Chong & Treisman, 2005) and it is likely to be dependent from later stages in visual processing.

Interestingly, the unmanipulated face received a high mean attractiveness rating, which was only second to that of the centered nose. This was somewhat unexpected, since also the picture where the mouth was centered would make the face appear more symmetrical and more average. Yet, our symmetry manipulations may both have increased and decreased the size of the mouth and the nose depending on the size ratio of the two sides of the manipulated character (Rhodes, 2006). As, for example, a large mouth can be considered masculine and a small mouth feminine (Etcoff, 1999), our manipulations may thus have influenced attractiveness by introducing feminine characteristics in a masculine face, or vice versa. Consequently, the attractiveness of the unmanipulated picture may be high because the original proportions of the nose and mouth are not distorted and not too small or too large as the effect of manipulations.

Conclusions

The association between attractiveness and nose tip position, but not of mouth position, suggests a bias in perception of the two traits which might be mirrored in the common occurrence of rhinoplasty, mainly occurring among young unmarried women and men (Babuccu et al., 2003). Attractiveness has in both sexes been found to be one of the main criteria of mate choice (Speed & Gangestad, 1997) and facial traits showing a strong impact on attractiveness ratings seem ultimately related to either health and reproductive potential of an individual or they may be by-products of how the brain processes information (Rhodes, 2006). Our results correspond with those of Neby & Folstad (2013) and suggest that information-processing mechanisms may have been important for the evolution of the human nose. That is, the brain may process information about facial symmetry and averageness heavily relying on the centrality of nose-tip position (although we cannot exclude that other nose abnormalities, e.g., of the ridge of the nose, might have equal or stronger effects). Consequently, the protruding human nose may have evolved as a signal of underlying qualities of potential mates or rivals. Although the handicap principle may account for such evolution, we believe it evolved as an amplifier, where the average benefit to signalers has been positive, even though low-quality individuals were forced to provide information of their poor quality.

After starting this work, the authors have become extremely conscious about noses, and that might also be the case with Erlend J. Jensen, Geir Rudolfsen, Lars Figenschou, Ismael Galván and Claus Wedekind who are all thanked for their constructive criticisms. Thanks also to the anonymous student who approved the publication of the facial images used and to an anonymous referee, who suggested to test whether males judge other males with injured noses as intimidating, signalling willingness to engage in fights.

Additional Information and Declarations

Competing Interests

Author Contributions

Human Ethics

Nigel Yoccoz is an Academic Editor for PeerJ.

Åse Kristine Rognmo Mikalsen conceived and designed the experiments, performed the experiments, analyzed the data, contributed materials tools, wrote the paper, prepared figures, reviewed drafts of the paper.

Ivar Folstad conceived and designed the experiments, wrote the paper, reviewed drafts of the paper, initiated the study and supervised Åse during her master studies.

Nigel Gilles Yoccoz conceived and designed the experiments, analyzed the data, contributed materials tools, wrote the paper, reviewed drafts of the paper.

Bruno Laeng conceived and designed the experiments, performed the experiments, contributed materials tools, wrote the paper, reviewed drafts of the paper.

The following information was supplied relating to ethical approvals (i.e., approving body and any reference numbers):

We studied facial pictures from students that had approved such use at the Department of Psychology, University of Tromsø, Norway. The manuscript include pictures of one student, as an example of the pictures used. The student on these pictures has approved publication of the pictures.

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
