# Peer review of "The spectacular human nose: an amplifier of individual quality?"

_PeerJ, doi:10.7717/peerj.357_

## Round 0.1 · original submission · Minor Revisions

Both referees liked this manuscript and I agree with them. However, they also made a number of suggestions as to further improve the clarity of the manuscript. These points should be easily addressed in a revised version.

·

Basic reporting

Mikalsen and colleagues asked 80 male and female students to rate manipulated portrait photos. They found that facial attractiveness depends on how much nose tips are centered. The authors discuss their results in the context of signaling theory. I found the study well performed and the paper generally well written.
One minor point: The description of the scaling of attractiveness may potentially be misleading (line 149). I would suggest to write “… participants … had to rank all pictures for attractiveness” instead of “… participants … had to rank all pictures for attractiveness on a 7-point scale” (in order to avoid the misunderstanding that attractiveness was scored on a scale from 1 to 7).

Experimental design

The MS describes original primary research that is based on a straight-forward experimental design and that meets high scientific standards.

Validity of the findings

I generally find the conclusions appropriate and speculations in the Discussion well marked and useful. However, the present version seems to emphasize that manipulations of the nose tips create stronger effects on attractiveness than manipulations of the mouth (see Abstract and first sentence of the Discussion). They authors actually use somewhat careful wording here, but I still found their description potentially misleading, because (i) effects of manipulations of the nose were not directly tested against effects of manipulations of the mouth, and (ii) a statistically non-significant effect (manipulations of the mouth) cannot generally be interpreted as no effect (statistical power?).

Reviewer 2 ·

Basic reporting

The authors use smiling faces in their stimuli. This is a reasonable as it is a pro-social expression involved in interpersonal attraction. However, compared to neutral facial expressions, a posed open smile is judged as significantly more attractive, kind, sympathetic, ambitious and intelligent (Otta, Abrosio, & Hoshino, 1996). Thus, the results may have differed had smiling faces not been used.

Otta, E., Abrosio, F. F. E., & Hoshino, R. L. (1996). Reading a smiling face: Messages conveyed by various forms of smiling. Perceptual and Motor Skills, 82, 1111–1121.

Experimental design

The concordance in ratings between the sexes and the sex of the target stimuli suggest a domain general effect for the manipulations. Clearly showing a broken nose gives the face an asymmetrical appearance, which is why the mouth symmetry manipulation is very important to this study. I think the authors handled this well in the discussion (lines 218-242).

Validity of the findings

The study is straightforward and the methods clearly explained. The statistical procedures and reporting of the results are also clearly explained and easy to follow. The Figures are useful and demonstrate how the manipulations were undertaken and the direction of the main findings. My only suggestion is that separate panels be included for male faces and female faces possibly split by the participant’s sex. I realise there were no statistically significant effects for the sex of the face or the sex of the respondents, but it might be interesting to see those data nonetheless.

Additional comments

Thank you for the opportunity to review this paper. The authors assert that the human nose is sexually selected as a signal amplifier of quality. Here quality is defined as a handicap trait, as the nose is a frequent target of blows during fights. To test whether nasal shape is sexually selected the authors manipulated the angle of nose in both male and female faces to mimic breakage from a fight. They also manipulated the angle of the mouth to assure that the nose manipulations do not simply become an artefact of making faces look asymmetrical. Male and female participants then rated these images for attractiveness. Stimuli with the centred nose were most attractive followed by the un-manipulated nose and un-manipulated mouth, suggesting that symmetry of the human nose enhances the sexual attractiveness of both male and female faces.

Overall, the findings of this study are interesting and raise some interesting theoretical perspectives regarding the evolution of human facial shape by sexual selection. I have the following comments and suggestions:

1. Participants rated stimuli for attractiveness only and there could be effects of the type of relationship (short-term or long-term) on the attractiveness of the stimuli. For example, men with facial scars are judged to be more attractive for short-term than long-term relationships (Burriss, Rowland, & Little, 2009). Thus, a broken nose could be seen as a risky or dominant male who could be more attractive when considering a short-term than a long-term relationship.

2. To bolster the discussion where the authors mention hormonal effects on facial shape (Lines 215-217), the authors could include some discussion of androgens in craniofacial development. For example, Verdonck et al., (1999) showed that masculine facial shape (e.g. mandible length, upper face height) increased in boys with delayed puberty when administered with a low does of testosterone.

3. The authors state (Lines 281-282) that “the protruding nose may have evolved as a signal of underlying qualities of potential mates and rivals”.

I agree and the authors test whether attractiveness is affected by the appearance of a broken nose. However, if the size and prominence of the human nose is sexually selected as a handicap signal, it may have been interesting to ask whether men perceive other men (more than women) as stronger, better fighters or more physically aggressive with a broken nose. This way one could test whether males judge other males with injured noses as intimidating as a signal that they are more willing to engage in fights. As an example, Stirrat et al., (2012) showed that men with more masculine shaped faces (measured using the width-to-height ratio) were less likely to die from contact violence than men with less masculine faces. Further, the human beard has been shown to amplify aggressive facial expressions to men, but not necessarily attractiveness to women, possibly because it enhances the size of the face and the jaw (Dixson & Vasey 2012). The authors could discuss these studies in relation to future research on the role of intra-sexual signalling and the nose as a handicap signal.

I hope these comments were useful.

Additional References

Burriss, R. P., Rowland, H. M., & Little, A. C. (2009). Facial scarring enhances men’s attractiveness for short-term relationships. Personality and Individual Differences, 46, 213-217.

Dixson, B. J., & Vasey, P. L. (2012). Beards augment perceptions of men's age, social status, and aggressiveness, but not attractiveness. Behavioral Ecology, 23, 481-490.

Stirrat, M., Stulp, G., & Pollet, T. V. (2012). Male facial width is associated with death by contact violence: narrow-faced males are more likely to die from contact violence. Evolution and Human Behavior, 33, 551-556.

Verdonck, A., Gaethofs, M., Carels, C., & de Zegher, F. (1999). Effect of low-dose testosterone treatment on craniofacial growth in boys with delayed puberty. The European Journal of Orthodontics, 21, 137-143.

---

## Round 0.2 · accepted · Accept

Thank you for this excellent article.